# Penifuranone A: A Novel Alkaloid from the Mangrove Endophytic Fungus *Penicillium crustosum* SCNU-F0006

**DOI:** 10.3390/ijms25095032

**Published:** 2024-05-05

**Authors:** Hao Jia, Li Wu, Rongrong Liu, Jialin Li, Lingling Liu, Chen Chen, Junsen Li, Kai Zhang, Junjiang Liao, Yuhua Long

**Affiliations:** Guangzhou Key Laboratory of Analytical Chemistry for Biomedicine, School of Chemistry, South China Normal University, Guangzhou 510006, China; haojia@m.scnu.edu.cn (H.J.); wuli@m.scnu.edu.cn (L.W.); rongrongliu@m.scnu.edu.cn (R.L.); jialinli@m.scnu.edu.cn (J.L.); linglingliu600@m.scnu.edu.cn (L.L.); chenchen2021@m.scnu.edu.cn (C.C.); junsenli@m.scnu.edu.cn (J.L.); zhangkai2021@m.scnu.edu.cn (K.Z.); junjiangliao0712@m.scnu.edu.cn (J.L.)

**Keywords:** mangrove fungus, alkaloid, antimicrobial activity, anti-inflammatory activity

## Abstract

One previously undescribed alkaloid, named penifuranone A (**1**), and three known compounds (**2**–**4**) were isolated from the mangrove endophytic fungus *Penicillium crustosum* SCNU-F0006. The structure of the new alkaloid (**1**) was elucidated based on extensive spectroscopic data analysis and single-crystal X-ray diffraction analysis. Four natural isolates and one new synthetic derivative of penifuranone A, compound **1a**, were screened for their antimicrobial, antioxidant, and anti-inflammatory activities. Bioassays revealed that penifuranone A (**1**) exhibited strong anti-inflammatory activity in vitro by inhibiting nitric oxide (NO) production in lipopolysaccharide-activated RAW264.7 cells with an IC_50_ value of 42.2 μM. The docking study revealed that compound **1** exhibited an ideal fit within the active site of the murine inducible nitric oxide synthase (iNOS), establishing characteristic hydrogen bonds.

## 1. Introduction

Natural products remain an important source of lead compounds and innovative medicines. From 1981 to 2019, 49.5% of all drugs approved by FDA were derived from natural products or their derivatives [1]. Marine microorganisms are considered to be an important source of natural products with marvelous structures and numerous biological activities [2]. Mangrove forests are salt-tolerant woody plants that form coastal intertidal and fragile ecosystems at the intersection between land and water [3]. The intricate root system of mangroves shelters abundant microorganisms [4]. Among the mangrove microorganisms, mangrove fungi are extremely interesting and considered as an important resource of valuable natural products with chemical structures and biological diversity [5,6]. Some of the key secondary metabolites derived from mangrove endophytic fungi include alkaloids [7,8], terpenoids [9], polyketides [10,11], peptides [12,13], etc. Alkaloids constitute one of the largest categories among these secondary metabolites and have a wide range of biological activities, such as anticancer [14,15], anti-neurodegenerative [16], anti-diabetic [17], anti-inflammatory [18], and antimicrobial activities [19], among others. Exploiting new alkaloid natural products is of great importance for drug development.

At present, microbial infection is a serious threat to human health [20,21]. The rapid increase in the bacterial resistance to the available antibiotics is a serious problem for the treatment of various infections globally. In 2019, 4.95 million people died globally due to drug-resistant infections [22]. Not only that, but persistent infection with pathogenic microorganisms can cause inflammatory response in various parts of the body [23,24]. Chronic or sustained inflammatory responses can cause tissue damage; potentially trigger acute inflammatory diseases such as acute gastroenteritis, bacteremia, toxemia, and acute renal disease, as well as chronic diseases such as chronic nephritis, chronic hepatitis, chronic bronchitis, and cancer; or even result in death [25,26]. As a result, the pursuit of innovative drugs which exhibit enhanced efficacy and fewer side effects and effectively manage inflammatory diseases and microbial infection is crucial [27]. 

Tetronic acids, which are common fragments among natural products originating from a variety of marine and terrestrial species [28,29,30], are vital in the fields of biology and chemistry [31]. Studies on this kind of compound have experienced a renaissance due to these products’ wide rang of biological activities, such as their anti-inflammatory, anti-tumor [32], antimicrobial activities [33], and so on. In addition, tetronic acids are also key pharmacophores of pesticides [34]. They play an important role in controlling agricultural pathogenic microorganisms and pests, as well as in regulating plant growth, and have a far-reaching influence on searching for environment-benign agrochemicals for humankind.

During our ongoing search for new bioactive natural products from mangrove endophytic fungi, a novel alkaloid, penifuranone A (**1**), featuring a fantastic piperidine fused tetronic acid ring system and three known compounds were isolated and identified from *Penicillium crustosum* SCNU-F0006 (Figure 1). Furthermore, a new derivative of penifuranone A, compound **1a**, was semi-synthesized to expand the new active compound library for bioassay screening. Compounds **1** and **2** were tested for their anti-inflammatory activity. And all the obtained compounds were used in antimicrobial and antioxidant activity biological assays. The results disclosed that penifuranone A (**1**) showed strong anti-inflammatory activity. Thus, the isolation, structure elucidation, and bioactivity evaluation of **1**–**4** and **1a** are described herein.

## 2. Results and Discussion

### 2.1. Structure of Compound ***1***

Compound **1** was obtained as a colorless crystal with the molecular formula of C_13_H_15_ NO_5_, with 7 degrees of unsaturation deduced from its HRESIMS ion peak in negative mode at *m/z* 264.0877 [M − H]^−^ (calcd for C_13_H_14_NO_5_, 264.0876) (Appendix A). The ^1^H NMR of compound **1,** along with the HSQC data (Table 1, Appendix A), indicated the presence of two methyls [1.3 (3H, d, *J* = 6.66 Hz, H-13), 1.3 (3H, d, *J* = 6.60 Hz, H-14)], two methylenes [3.2 (2H, m, H-2), 1.7 (2H, m, H-3)], three methines [3.3 (1H, t, *J* = 5.76 Hz, H-4), 4.7 (1H, q, *J* = 6.66 Hz, H-11), 4.6 (1H, q, *J* = 6.60 Hz, H-7)], one hydroxy proton [11.5 (1H, s, 12-OH)], and one secondary amine proton [7.4 (1H, s, 1-HN)]. Analysis of the ^13^C NMR spectrum, associated with the HSQC spectrum, showed 13 carbon resonances, which were assigned to two carbonyl groups [172.6 (C-10), 171.26 (C-8)], four olefinic carbons [167.7(C-6), 88.5(C-5), 100.5(C-9), 176.6(C-12)], two oxygenated tertiary carbons [71.8(C-7), 72.8(C-11)], two methyls [18.9(C-14), 18.13(C-13)], two methylenes [39.8(C-2), 26.3(C-3)], and one methine [23.3(C-4)]. Detailed analysis of the ^1^H-^1^H COSY (Figure 2 and Appendix A), HSQC (Appendix A), and HMBC spectra data (Figure 2 and Appendix A) allowed for the assembly of the planar structure of **1**. The HMBC correlations from H-7 to C-5, C-6, and C-8 established the *γ*-butyrolactone (ring A). The ^1^H-^1^H COSY correlations of spin system H-1/H_2_-2/H_2_-3/H-4, as well as the HMBC correlations from H-1 to C-3, C-5, and C-7; from H-4 to C-2 and C-6; from H-7 to C-5 and C-6; and from H-14 to C-6, indicated the existence of a tetronic acid (ring A) fused with a 2-piperideine (ring B) formed between C-5 and C-6 (Figure 2). The HMBC correlations from H-11 to C-9, C-10, and C-12 established another tetronic acid (ring C). The key HMBC correlations from H-4 to C-9, C-10, and C-12 suggested the connectivity of ring A to ring C via C-4. Two methyl groups were connected to C-7 and C-11 by the ^1^H-^1^H COSY correlations of spin systems H_3_-13/H-11 and H_3_-14/H-7, coupled with the HMBC associations from H_3_-13 to C-11 and C-12 and from H_3_-14 to C-6 and C-7. So far, the planar structure of **1** has been established. The relative configurations of **1** were partly identified by the NOESY spectrum (Figure 2 and Appendix A). The NOESY correlations of H-4/H_3_-13/H_3_-14 implied that these protons were on the same side, as shown in Figure 2. A suitable single crystal of **1** was obtained from a methanol and dichloromethane system (2:1). The absolute configuration of **1** was unambiguously determined as 4*R*, 7*R*, 11*R* by a single-crystal X-ray diffraction experiment using Cu K*α* radiation with the Flack parameter of [0.09(10)] (Figure 3). In addition, the structures of (+) - (5*R*, 5′*R*)-3, 3′-methylenebistetronic acid (**2**) [35], viridicatin (**3**) [36], and viridicatol (**4**) [37] were determined by comparing their NMR data with those reported in the literature.

### 2.2. Semi-Synthesis of Compound ***1a***

Due to the fact that the hexadecanoyl group always benefits the antimicrobial activity, we semi-synthesized a hexadecanoyl derivative of compound **1** with a limited available amount of the isolated compound **1** (8 mg) [31,38]. The synthetic route for compound **1a** is depicted in Figure 1. One step reaction of penifuranone A (**1**) with hexadecanoyl chloride gave **1a** in up to 90% yields. The structure of target compound **1a** was confirmed by extensive spectroscopic methods, including HRESIMS (Appendix A), ^1^H NMR (Appendix A), and ^13^C NMR (Appendix A).

### 2.3. Antimicrobial Assay

The antimicrobial activities of compounds **1**–**4** and **1a** were investigated. As shown in Table 2, all the five compounds displayed modest inhibitory activities against *Fusarium oxysporum* and *Penicillium italicum*, with MIC values of 0.25 mg/mL, except for **1a**, with an MIC of 0.125 mg/mL to *Penicillium italicum*. The results from Table 3 indicate that compounds **1**–**4** and **1a** can inhibit the growth of *Pseudomonas aeruginosa*, with MIC values ranging from 0.5 mg/mL to 0.125 mg/mL. The SAR analysis indicated that the antimicrobial potency of compound **1a** against *Penicillium italicum* and *Pseudomonas aeruginosa* was higher than that of compound **1**. As for the other tested strains, these compounds showed no obvious inhibitory activity.

### 2.4. Anti-Inflammatory Assay

Initially, we evaluated the cytotoxic effects of all the compounds on RAW 264.7 cells (Table 4). It was observed that compounds **1** and **2** showed low cytotoxicity, with IC_50_ values above 50 μM, whereas the IC_50_ values for the remaining compounds were below this threshold. Consequently, we focused our subsequent anti-inflammatory activity assessments solely on compounds **1** and **2**. The anti-inflammatory activities of compounds **1** and **2** were evaluated using lipopolysaccharide (LPS)-stimulated RAW 264.7 cells. Notably, compounds **1** and **2** showed significant inhibitory activity on NO production, with IC_50_ values of 42.22 and 37.01 μM, respectively (Dexamethasone as a control, IC_50_ = 58.21 μM). In the MTT assay, none of the compounds except for the three compounds of **1a**, **3**, and **4** exhibited significant cytotoxic effects on the RAW264.7 cells (IC_50_ < 50 μM). 

### 2.5. DPPH Scavenging Assay

Compounds **1**–**4** and **1a** were evaluated for their antioxidant activity using the DPPH scavenging assay (Table 5). Compounds **1** and **2** showed moderate radical scavenging activity, with IC_50_ values of 180.2 μM and 120.5 μM, respectively, while the other compounds exhibited no inhibitory effect at a concentration of 200 μM. Vitamin C was used as a control (IC_50_ = 68.4 μM).

### 2.6. Molecular Docking Studies

Modulating the excessive production of nitric oxide is commonly pursued by targeting the inducible nitric oxide synthase (iNOS) enzyme to suppress its expression or activity [39,40]. To explore the suppressive action of compounds **1** and **2** on nitric oxide (NO) synthesis, the study focused on the interplay and interaction pattern between compounds **1**–**2** and the inducible nitric oxide synthase (iNOS) enzyme, as depicted in the Protein Data Bank entry 3E6T [41]; thus, a molecular docking study was carried out using AUTODOCK 4.2.6 modeling software. The docking procedure was validated by the docking of the ligand indomethacin (positive drug) in the active site of iNOS and a root-mean-square deviation (RMSD) of 0.12 Å to the X-ray structure. The results showed that the binding energy of compound **1** (−6.32 Kcal/mol) and compound **2** (−5.58 Kcal/mol) was comparable to that of the positive drug indomethacin (−7.45 Kcal/mol). Further observations showed that indomethacin formed a hydrogen bond with the key amino acid residue GLU-371, two hydrogen bonds with the amino acid residue ARG-260, and a hydrogen bond with GLN-257 in the iNOS active pocket (Figure 4A) [42,43]. Compound **1** formed a hydrogen bond with the key amino acid residue ARG-382 through the ester group, three hydrogen bonds with the residue ASP-379, and one hydrogen bond with the residue HEM-901 in the iNOS active pocket (Figure 4B). Compound **2** formed three hydrogen bonds with the key amino acid residue GLU-371 through two hydroxyl groups, one hydrogen bond with the residue 1A2-905, and one hydrogen bond with the residue HEM-901 in the iNOS active pocket (Figure 4C). As a result, compounds **1** and **2** exhibited comparable inhibitory activity on iNOS to the positive control drug indomethacin (Table 4).

## 3. Experimental Section

### 3.1. General Experimental Procedures

HRESIMS data were obtained on a Finnigan LTQ-Orbitrap Elite (ThermoFisher Scientific, Waltham, MA, USA). Nuclear Magnetic Resonance (NMR) spectra data were obtained using the Bruker AVANCE NEO 600 MHz spectrometer (Bruker BioSpin, Rheinstetten, Germany) using TMS as an internal reference at room temperature. Optical rotations were determined using an Anton Paar (MCP 300) polarimeter at 25 °C. UV spectra were measured on a UV-2700 (Shimadzu, Kyoto, Japan) spectrophotometer [*λ*_max_ (log *ε*) in nm] in MeOH solution. IR spectra were measured on a Nicolet Nexus 670 spectrophotometer in KBr discs. Sephadex LH-20 (25–100 μm; GE Healthcare Bio-Sciences AB, Stockholm, Sweden) and silica gel (200–300 mesh; Qingdao Marine Chemical Factory, Qingdao, China) were used for column chromatography. Thin-layer chromatography (TLC) analyses were conducted with precoated silica gel GF254 plates (Qingdao Marine Chemical Ltd., Qingdao, China). TLC silica gel 60 RP-18 F254s (Merck KGaA, 64271 Darmstadt Germany) was used. X-ray crystallographic analysis was performed on an Agilent Gemini Ultra diffractometer (Cu K*α* radiation). Chiral HPLC analysis was carried out by photodiode array (PDA) analysis using a SHIMADZU Prominence LC-20A HPLC system (SHIMADZU, Kyoto, Japan) with a Phenomenex column (Gemini, 250 × 4.6 mm, C18, 5 µm; Phenomenex, Torrance, CA, USA).

### 3.2. Plant Material

*Acanthus ilicifolius* L. mangrove plants were collected from the Yangjiang Mangrove Nature Reserve in Guangdong province, China (E 111°48′06″ to 112°11′59″, N 21°33′21″ to 21°40′18″) (Figure 5). 

The management personnel of the mangrove nature reserve classified and identified various mangrove plants (Mangrove Management Center in Hailing District, Yangjiang City, China). Morphological Features: the leaves of *Acanthus ilicifolius* L. are opposite, pinnately cleft or shallowly lobed, and often have sharply spiny teeth. In florescence and Flowers: The terminal spike-like inflorescence of *Acanthus ilicifolius* L. is borne at the top, with large bracts that commonly have spines along the edges and smaller bracteoles that may be absent. The corolla is bilabiate, with the upper lip being extremely reduced and appearing almost like a single lip, while the lower lip is large, spreading, and trifid. The calyx is lobed 4–5 times, and the corolla tube is very short, nearly spherical, or has a very narrow cylindrical shape. Fruit and Seeds: The capsule of *Acanthus ilicifolius* L. is elliptical, laterally compressed, lustrous, and chestnut brown in color, containing four seeds. The seeds are laterally compressed, nearly round, or broadly ovate and possess a funicular hook.

### 3.3. Fungal Material

The fungal strain *Penicillium crustosum* SCNU-F0006 was isolated from the fresh stems *Acanthus ilicifolius* L. mangrove plants collected from the Yangjiang Mangrove Nature Reserve in Guangdong province, China (Figure 5). The management personnel of the mangrove nature reserve classified and identified various mangrove plants. On 9 October, 2021, fresh samples collected from the mangrove wetland conservation area were first cleaned with sterile water, with the plant’s roots, stems, leaves, flowers, fruits, and seeds being separated and placed individually. Sterilized forceps were used to sequentially transfer the plant tissues through a 3% sodium hypochlorite solution and a 75% ethanol solution for disinfection, followed by rinsing with sterile water to remove residual solvents. The stems of the plant known as *Acanthus ilicifolius* L. were then cut into regular small pieces (approximately 0.2 × 0.6 cm) and transferred onto a Bengal Rose Agar plate that had been sterilized with high-pressure steam. After sealing the plate, it was incubated at room temperature in a sterile environment. After 2–3 days, as colonies emerged, a sterile loop that had been flamed with an alcohol lamp was used to take sections of mycelium from different colonies and transfer them to Potato Dextrose Agar (PDA) medium that had been sterilized with high-pressure steam. The process was repeated until a single pure colony was present in each plate, indicating the completion of purification. The purified strains were then transferred to sterile PDA slant tubes for storage in a refrigerator for future use. The fungus was identified by using ITS gene sequencing (seq: ACCCTCCGTTTAGGGGAACTGCGGAAGGATCATTACCGAGTGAGGGCCCTCTGG GTCCAACCTCCCACCCGTGTTTATTTTACCTTGTTGCTTCGGCGGGCCCGCCTTAA CTGGCCGCCGGGGGGCTTACGCCCCCGGGCCCGCGCCCGCCGAAGACACCCTCG AACTCTGTCTGAAGATTGAAGTCTGAGTGAAAATATAAATTATTTAAAACTTTCA ACAACGGATCTCTTGGTTCCGGCATCGATGAAGAACGCAGCGAAATGCGATACG TAATGTGAATTGCAAATTCAGTGAATCATCGAGTCTTTGAACGCACATTGCGCCC CCTGGTATTCCGGGGGGCATGCCTGTCCGAGCGTCATTGCTGCCCTCAAGCCCGG CTTGTGTGTTGGGCCCCGTCCCCCGATCTCCGGGGGACGGGCCCGAAAGGCAGC GGCGGCACCGCGTCCGGTCCTCGAGCGTATGGGGCTTTGTCACCCGCTCTGTAGG CCCGGCCGGCGCTTGCCGATCAACCCAAATTTTTATCCAGGTTGACCTCGGATCA GGTAGGGATACCCGCTGAACTTAAGCATATCAATAAGGCGGAGGAAA). The sequence data have been stored in Gen Bank with accession no. MH345907. The BLAST result showed that the sequence was most similar (99%) to the sequence of *Penicillium crustosum.* A voucher strain was deposited in School of Chemistry, South China Normal University, Guangzhou, China, with the access code SCNU-F0006.

### 3.4. Fermentation and Isolation

*Penicillium crustosum* SCNU-F0006 was fermented on solid autoclaved rice medium using 100 1 L Erlenmeyer flasks, each of which contained 40 g rice and 40 mL 0.3% sea salt, and cultured at room temperature under static conditions for 28 days. The solid rice medium and mycelia were extracted using methanol and ethyl acetate, each for three times, with each extraction lasting for 24 h. The organic solvents were evaporated under reduced pressure, and then a crude extract was obtained. The crude extract was subjected to solid–liquid separation using a vacuum filtration funnel. The solid residue was further soaked with ethyl acetate, and the liquid portion was extracted with ethyl acetate as well. Ultimately, an ethyl acetate extract weighing 78.0 g was obtained. The extract was isolated by column chromatography over silica gel eluting with a gradient of petroleum ether: ethyl acetate from 1:0 to 0:1 to afford five fractions (Fractions 1–5: Fraction 1 (15.0 g), Fraction 2 (23.6 g), Fraction 3 (14.3 g), Fraction 4 (8.5 g), Fraction 5 (12.9 g)). Fraction 1 (220.0 mg) and Fraction 2 (220.0 mg) were applied to Sephadex LH-20 CC and eluted with CH_2_Cl_2_/MeOH (1:1) to afford three sub-fractions (SFrs. 1.1–1.3). SFr. 1.2 (125.2 mg) was applied to column chromatography over silica gel, eluting with CH_2_Cl_2_/MeOH (200:1) and MeOH/H_2_O (60:40) to obtain compound **2** (20.5 mg). SFr. 1.3 (76.3 mg) was applied to column chromatography over silica gel. eluting with CH_2_Cl_2_/MeOH (160:1), petroleum ether/ethyl acetate (*v*/*v*, 9:1), CH_2_Cl_2_/MeOH (180:1), and MeOH/H_2_O (60:40) to obtain compound **3** (40.2 mg). Fraction 3 (160.0 mg) was applied to column chromatography over silica gel, eluting with CH_2_Cl_2_/MeOH (100:1), and then further purified by Sephadex LH-20 CC, eluting with CH_2_Cl_2_/MeOH (1:1) and MeOH/H_2_O (60:40) to obtain compound **4** (80.0 mg). Fraction 4 (136.0 mg) was applied to Sephadex LH-20 CC and eluted with CH_2_Cl_2_/MeOH (1:1) to afford three sub-fractions (SFrs. 4.1–4.3). SFrs. 4.2 (42.3 mg) was applied to column chromatography over silica gel, eluting with CH_2_Cl_2_/MeOH (100:1), and then further purified by Sephadex LH-20 CC, eluting with MeOH, further purified by column chromatography over silica gel, eluting with petroleum ether/ethyl acetate (*v*/*v*, 8:1), then using HPLC with Phenomenex column (Gemini, MeOH/H_2_O *v*/*v*, 60:40, 250 × 4.6 mm), to give compound **1** (8 mg, t_R_ = 10.2 min) (8 mg) (Appendix A). The yield of the new compound **1** was 0.01%, and the purity, as analyzed by HPLC, was 98.74% (Appendix A). The yields of compounds **2**–**4** were 0.026%, 0.052%, and 0.10%, respectively, and the purity was analyzed by TLC, as shown in Appendix A.

### 3.5. Infrared Spectroscopy

For infrared spectroscopy, the following step-by-step protocol was used: Firstly, use an electronic balance to weigh out the required amounts of the sample to be tested: 1 mg compound and 120 mg of spectroscopic-grade potassium bromide (KBr) that has been dried in an oven. Then, place the dried samples and KBr into an agate mortar and grind them for 2–3 min until a uniform powder is formed. Transfer the ground powder into a pellet press mold. Utilize a specially designed mold to ensure that the sample powder is evenly distributed, with the center slightly higher than the edges. Place the mold containing the sample into the press and apply pressure (typically between 10 and 20 MPa) for 40 s. During the pressing process, connect to a vacuum pump to evacuate air and moisture from both the sample and KBr powder. After pressing, slowly release the pressure and remove the pressed KBr pellet. Place the pressed pellet into an infrared spectrometer for spectral scanning. To eliminate the infrared absorption peaks of the KBr carrier itself, it is usually necessary to first scan a pure KBr pellet as a background. Based on the obtained infrared spectrum, analyze the characteristic absorption peaks of the sample to determine its chemical structure.

### 3.6. LC-MS Parameters

LC-MS (ThermoFisher Scientific, Waltham, MA, USA) and a C18 column (ThermoFisher Scientific-packed Hypersil GOLD, 1.9 μm, 2.1 × 100 mm) were used to analyze the compounds. A total of 20 μL of sample (1 mg compound was dissolved in 1 mL MeOH) was injected and eluted with an isocratic. The detailed procedure for isocratic elution involves using a liquid phase flow rate of 1 mL/min with the following mobile phases: Phase A consisting of 95% water, 5% acetonitrile, 0.01% formic acid, and 2.5 mM ammonium formate, and Phase B consisting of 95% acetonitrile, 5% water, 0.01% formic acid, and 2.5 mM ammonium formate. The ion source used was an ESI (Electrospray Ionization) source, operated in either positive or negative ion mode. The mass spectrometer conditions were set as a spray voltage of 3.0 kV and ion transfer tube temperature of 300 °C.

### 3.7. Synthetic Method and Spectral and Physical Data of Compounds ***1*** and ***1a***

Compound **1**: colorless crystals; m.p. 156.7–158.2 °C. [α${]}_{D}^{25}$ +88 (c = 0.1, MeOH); UV(MeOH) *λ*_max_ (log *ε*) 229.53(1.01), 267.87(1.46) (Appendix A); IR (KBr) *ν*_max_ 3359.78, 3277.07, 3098.59, 2959.29, 2824.44, 2711.35, 1723.66, 1678.97, 1622.43, 1563.15, 1273.14, 1053.35, 1002.27, 958.50, 769.72 cm^−1^ (Appendix A); ^1^H (600 MHz, DMSO-*d*_6_) and ^13^C (150 MHz, DMSO-*d*_6_) NMR data (see Table 1); HRESIMS: *m/z* 264.0877 [M − H]^−^ (calcd for C_13_ H_14_ O_5_ N, 264.0876).

Compound **1a**: To a rounded bottom flask, 5 mg of penifuranone A (**1**) in 5 mL of anhydrous dichloromethane, 1 eq hexadecanoyl chloride, and 0.2 mL of triethylamine were added in. The reaction mixture was stirred at 0 °C for 15 min and then maintained at room temperature (rt) for 6 h and monitored by TLC until penifuranone A (**1**) disappeared, indicating the reaction endpoint. The reaction was quenched by the addition of saturated salt water and extract with anhydrous dichloromethane. Then, we dried the reaction product over anhydrous Na_2_SO_4_ and evaporated the solvent. The obtained raw product was further purified by column chromatography with an eluent of dichloromethane/methanol (0→20%) (Appendix A). White solid; m.p. 76.6–77.2 °C. [α${]}_{D}^{25}$ +34 (c = 0.1, MeOH); UV(MeOH) *λ*_max_ (log *ε*) 223.85(3.01), 265.21(4.88) (Appendix A); IR (KBr) *ν*_max_ 2915.64, 2847.24, 2680.34, 1758.32, 1654.35, 1465.57, 1415.41, 1264.02, 1194.71, 939.35, 936.61, 801.65, 715.91 cm^−1^ (Appendix A); ^1^H NMR (600 MHz, CDCl_3_) *δ*_H_ 5.2 (q, 1H, *J* = 6.00 Hz), 4.1 (q, 2H, *J* = 6.00 Hz), 3.6(t, 1H, *J* = 6.00 Hz,), 3.4 (m, 1H), 3.8 (m, 1H), 2.5–2.4 (m, 2H), 2.3 (d, 2H, *J* = 6.00 Hz), 2.3 (s, 1H), 2.1 (m, 1H), 1.9 (m, 1H), 1.6 (m, 6H), 1.4 (d, 3H), 1.4 (d, 3H), 1.4–1.2 (m, 44H, overlap), 0.9 (t, 6H, *J* = 6.00 Hz), ^13^C NMR (150 MHz, CDCl_3_) *δ*_C_ 178.93, 171.91, 171.00, 169.57, 168.77, 166.97, 115.86, 90.60, 74.77, 72.98, 40.09, 34.00, 33.93, 32.07, 29.88–29.54(overlap), 29.50, 29.39, 29.31, 29.21, 29.13, 25.35, 24.85, 24.67, 22.83, 19.02, 17.62, 14.26. HRESIMS: *m/z* 742.5615 [M + H]^+^ (calcd for C_45_H_76_NO_7_, 742.5616).

Compound **2**: (+) - (5*R*, 5′*R*) - 3,3′-methylenebistetronic acid. Colorless block crystals; HRESIMS (*m/z*): [M – H]^−^, calcd for C_11_H_12_O_6_, 239.0561; found, 239.0562; ^1^H NMR (CD_3_OD): *δ*_H_ 4.8 (q, *J* = 6.7 Hz), 3.0 (s), 1.0 (d, *J* = 6.7 Hz), 4.8 (q, *J* = 6.7 Hz), 1.0 (d, *J* = 6.7 Hz). ^13^C NMR (CD_3_OD): *δ* 179.6, 179.6, 177.9, 177.9, 97.8, 97.8, 76.4, 76.4, 18.0, 18.0, 14.2. 

Compound **3**: 3-hydroxy-4-(3-hydroxyphenyl)quinolin-2(1H)-one. White solid; HRESIMS (*m/z*): [M + H]^+^, calcd for C_15_H_12_NO_3_, 254.0431; found, 254.0432; ^1^H NMR (600 MHz, DMSO-*d*_6_) *δ*_H_ 12.2 (1H, s, OH), 9.5 (1H, s, OH), 9.0 (1H, s, NH), 7.3 (1H, dd, *J* = 8.0, 1.0), 7.3 (1H, ddd, *J* = 8.0, 8.0, 1.0), 7.3 (1H, dd, *J* = 8.0, 8.0), 7.1 (1H, dd, *J* = 8.0, 1.0), 7.1 (1H, ddd, *J* = 8.0, 8.0, 1.0), 6.8 (1H, m), 6.7 (1H, ddd, *J* = 8.5, 1.5, 1.0), 6.7 (1H, dd, *J* = 2.5, 1.5). ^13^C NMR (150 MHz, DMSO-*d*_6_) *δ*_C_ 158.3, 157.3, 142.2, 134.9, 133.1, 129.4, 126.4, 124.5, 124.1, 122.2, 120.9, 120.4, 116.7, 115.2, 114.6.

Compound **4**: 3-hydroxy-4-phenylquinolin-2(1*H*)-one. White solid; HRESIMS (*m/z*): [M + H]^+^, calcd for C_15_H_12_NO_2_, 238.1234; found, 238.1233; ^1^H NMR (600 MHz, DMSO-*d*_6_) *δ*_H_ 12.2 (1H, s, OH), 9.2 (1H, s, NH), 7.5 (1H, m), 7.5 (1H, m), 7.4 (1H, m), 7.3 (2H, m), 7.3 (1H, m), 7.3 (1H, m), 7.1 (1H,m), 7.0 (1H,m). ^13^C NMR (150 MHz, DMSO-*d*_6_) *δ*_C_ 158.28, 142.44, 133.76, 133.17, 129.85, 129.85, 128.35, 128.35, 127.67, 126.44, 124.31, 123.94, 122.14, 120.91, 115.26.

### 3.8. X-ray Crystal Data for Compound ***1***

Compound **1** was filtered through a microporous membrane, and a suitable single crystal was successfully grown using the slow evaporation method from a 2:1 mixed solvent system of methanol and dichloromethane. The crystal structure and absolute configuration of **1** were determined by using data collected at T = 100 K with Cu Kα radiation (*λ* = 0.0031 Å) on an Agilent Gemini Ultra diffractometer. The structure was solved by direct methods using SHELXS-9729 and refined using full-matrix least-squares difference Fourier techniques [44]. Hydrogen atoms bonded to carbons were placed on the geometrically ideal positions and refined using a riding model. Crystallographic data of **1** were saved in Cambridge Crystallographic Data Centre. (Deposition number: CCDC 2337713 for **1**) Copies of the data can be obtained, free of charge, upon application to the Director, CCDC, 12 Union Road, Cambridge CB2 1EZ, UK (fax: 44-(0)1223-336033; e-mail: deposit@ccdc.cam.ac.uk).

Crystal data of **1**: C_13_H_15_NO_5_, Mr = 283.27, mono-clinic, a = 9.4666(2) Å, b = 9.4666(2) Å, c = 11.3398(2) Å, *α* = 90.00, *β* = 108.934(2)°, *γ* = 90.00, V = 691.31(2) Å, space group P21, Z = 2, *D*_calc_ = 1.361 g/cm^3^, *μ* (Cu K*α*) = 0.919 mm^−1^, and F (000) = 300.0. Crystal size: 0.15 × 0.13 × 0.11 mm^3^. Theta range for data collection: 4.9730° to 73.9220°. Reflections collected: 13024. Independent reflections: 2766 [*R*_int_ = 0.0396, *R*_sigma_ = 0.0257]. Goodness-of-fit on F2 = 1.107. Final R indices: R_1_ = 0.0319, *w*R_2_ = 0.0848. Final R indices (all data): *R*_1_ = 0.0345, *wR*_2_ = 0.0854. Flack parameter= −0.09(10). CCDC number: 2337713.

### 3.9. In Vitro Antifungal Activity

The bioactivity test screened the following four phytopathogenic microorganisms: *Fusarium oxysporum*; *Penicillium italicum*; *Banana Colletotrichum gloeosporioides*; and *Colletrichum litchi Trag*. The minimum inhibitory concentration (MIC) was determined using the broth dilution method, with a minor modification [45,46,47,48]. Fungi were removed from the refrigerator and cultured dynamically at a constant temperature in Potato Dextrose Broth at room temperature for 72 h. The fungal liquid was diluted with broth to 0.5 McFarland turbidity. The sample was prepared into an initial solution of 1 mg/mL with dimethyl sulfoxide (DMSO), and the concentrations of the 1st to 12th wells of the 96-well plate sample group were made by the double dilution method (mg/mL): 1, 0.5, 0.25, 0.125, and 0.0625. Each plate also contained a positive control (ciprofloxacin or carbendazim), as well as a sterile control and growth control (containing broth and fungal liquid, no compound). The plates were incubated at 35 °C for 72 h. The use of 10 μL of a solution with a concentration of 11 mg/mL of Esazurin added to each well of a 96-well plate, followed by incubation at 37 °C for 3 h, allowed for the observation of color changes to determine the minimum inhibitory concentration (MIC) values. A blue or purple color indicates that the inoculated bacteria have been completely inhibited, a purplish–red color suggests the partial inhibition of the fungal, and a pink color indicates fungal growth.

### 3.10. In Vitro Antibacterial Activity

The bioactivity test screened the following four human pathogenic microorganisms: *Pseudomonas aeruginosa* (ATCC 9027); Salmonella typhimurium (ATCC 6539); Escherichia coli (ATCC 25922); and *Staphylococcus aureus* (ATCC 12598). The minimum inhibitory concentration was determined using the broth dilution method [46,47], with a minor modification. Bacteria were removed from the refrigerator and cultured dynamically at a constant temperature in a Mueller–Hinton Broth at room temperature for 24 h. The bacterial liquid was diluted with broth to 0.5 McFarland turbidity. The sample was prepared into an initial solution of 1 mg/mL with dimethyl sulfoxide (DMSO), and the concentrations of the 1st to 12th wells of the 96-well plate sample group were made by the double dilution method (mg/mL): 1, 0.5, 0.25, 0.125, and 0.0625. Each plate also contained a positive control (ciprofloxacin or carbendazim), as well as a sterile control and growth control (containing broth and bacterial liquid, no compound). The plates were incubated at 35 °C for 24 h. The use of 10 μL of a solution with a concentration of 11 mg/mL of Esazurin added to each well of a 96-well plate, followed by incubation at 37 °C for 3 h, allowed for the observation of color changes to determine the minimum inhibitory concentration (MIC) values. A blue or purple color indicates that the inoculated bacteria have been completely inhibited, a purplish–red color suggests the partial inhibition of the bacteria, and a pink color indicates bacterial growth [48,49].

### 3.11. Anti-Inflammatory Assay

The MTT method was utilized to assess cell viability [25,26,27]. RAW 264.7 cells, at a concentration of 5 × 10^5^ cells/mL, were plated in 96-well plates and exposed to lipopolysaccharide (LPS, at 1 μg/mL) in the presence or absence of the test compounds for a duration of 24 h. Following this, 100 μL of a 2 mg/mL MTT solution was introduced into each well, and after a 4 h incubation, the resulting formazan precipitates were dissolved in 100 μL of DMSO. The optical density was then determined at 490 nm using a microplate reader (Multiskan GO, Thermo Scientific, Waltham, MA, USA). The results were presented as the average percentage of cell viability relative to the untreated control group.

Compounds **1**–**2** were evaluated for their inhibitory effects on LPS-stimulated NO production in RAW 264.7 cells using the Griess assay [45]. The cell line RAW 264.7, obtained from the Cell Bank of the Chinese Academy of Sciences (CBCAS) in Shanghai, China, was propagated in Dulbecco’s Modified Eagle Medium (DMEM) supplemented with 10% Fetal Bovine Serum (FBS). In the logarithmic growth phase, the cells were digested with trypsin and inoculated into 96-well plates (1 × 10^5^/well). After being cultured at 37 °C with 5% CO_2_ for 12 h, the cells were treated with DMEM containing different concentrations of compounds (0, 3.125, 6.25, 12.5, 25, 50, and 100 μM). Two hours later, the cells were treated with LPS at a concentration of **1** and **2** (1 μg/mL) and incubated under standard cell culture conditions for a period of 24 h. Fifty microliters of cell supernatant was mixed with an equal volume of Griess reagent (1% anhydrous *p*-aminobenzene sulfonic acid and 0.1% N-1-naphthalene ethylenediamine hydrochloride in 5% concentrated phosphoric acid). Absorbance was measured with a microplate reader at 540 nm. The IC_50_ values were calculated using the SPSS 25.0 statistical software analysis package based on the inhibition rate of each group.

### 3.12. DPPH Scavenging Assay

The assay for scavenging 1,1-Diphenyl-2-picrylhydrazyl radical 2,2-Diphenyl-1-(2,4,6-trinitrophenyl)hydrazyl (DPPH free radicals) was conducted following a previously established modified protocol [49]. The DPPH was dissolved in ethanol to prepare a 0.5 mmol/L DPPH ethanolic solution (10 mL), which was stored away from light. Subsequently, 100 μL of the DPPH solution was added to each well of a 96-well plate. The sample was prepared into an initial solution of 200 μM with ethanol, and the concentrations of the 1st to 12th wells of the 96-well plate sample group were made by the double dilution method (μM): 200,100, 50, 25, and 12.5 μM. The reaction was monitored by measuring the absorbance at 490 nm using a microplate reader (iMark, Bio-Rad, Hercules, CA, USA) after shaking the reaction for 30 min at 37 °C in the dark. The DPPH radical scavenging activity was determined by a comparison of the absorbance of the samples and a blank control. Vitamin C was used as a positive control at the same concentration as the other samples.

### 3.13. Molecular Docking

The AutoDock 4.2.6 software package includes an AutoDock tool that is utilized for conducting virtual docking simulations [39]. This is a prevalent approach in docking that provides the ligand with ample flexibility while preserving the target protein’s structural integrity. The X-ray crystal structure of iNOS (PDB ID: 3E6T) [41] was obtained from the RCSB protein database (PDB). Prior to the docking simulation, the original ligand and water molecules were removed from the crystal structure using PyMOL, and the resulting protein structure was exported in PDB format as ‘receptor.pdb’. The molecular structure was initially sketched in two dimensions using ChemDraw’s 2D 17.1 software. Subsequently, this 2D representation was transformed into a 3D configuration through the use of ChemDraw’s 3D 17.1 software. Finally, the 3D structure was saved as a file in the Protein Data Bank (PDB) format. Additionally, the molecular configuration was refined utilizing Gaussian software. Subsequently, both the protein and the ligand were formatted into PDBQT using the AutoDock suite to prepare them for the upcoming docking procedure. Focusing on the protein, the parameters of the grid box were set to 126 × 126 × 126 points, and the Lamarckian genetic algorithm was used to link the algorithm with 100 GA operations. Ultimately, the outcomes were examined and interpreted with the aid of PyMOL’s visualization capabilities. 

## 4. Conclusions

A chemical study of *Penicillium crustosum* SCNU-F0006 collected from Yangjiang Mangrove Nature Reserve led to the isolation and identification of one novel alkaloid, compound (**1**), together with three known compounds (**2**, **3**, and **4**). In addition, one novel derivative (**1a**) was synthesized from **1**. Compounds **1** and **2** were tested for their anti-inflammatory activity. And all compounds were tested for antioxidant and antimicrobial activities. The results displayed that compounds **1** and **2** showed a certain antioxidant activity and significant anti-inflammatory activity. Furthermore, compounds **1** and **1a** exhibited moderate antimicrobial activities against *Bacillus subtilis*, *Penicillium italicum*, and *Pseudomonas aeruginosa*.

## Data Availability

All data are available through corresponding author.

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
