# Peer review of "Penifuranone A: A Novel Alkaloid from the Mangrove Endophytic Fungus Penicillium crustosum SCNU-F0006"

_ijms, 2024, doi:10.3390/ijms25095032_

Round 1
Reviewer 1 Report
Comments and Suggestions for Authors
The manuscript entitled "Penifuranone A, a novel alkaloid from the mangrove endo- 2 phytic fungus Penicillium crustosum SCNU-F0006" discussed the isolation of compound, characterization, and their activity against different targets. This manuscript may be accepted after minor modifications.
1. Abstracts need to be more focused.
2. All the 1H and 13C NMR should be in the form of one decimal after only like 0.0 form.
3. Introduction is long and it should be concise.
Comments on the Quality of English Language
Grammatically it's ok but some spelling mistakes and some sentences need correction.
Author Response
Dear Reviewer:
Re: Manuscript ID: ijms-2966443.
Title: Penifuranone A, a novel alkaloid from the mangrove endophytic fungus Penicillium crustosum SCNU-F0006.
Thank you for arranging a timely review for our manuscript(ID: ijms-2966443). We have carefully evaluated your critical comments and thoughtful suggestions, responded to these suggestions point-by-point, and accordingly revised the manuscript. All changes made to the text are marked in red so that they may be easily identified. With regard to your comments and suggestions, our reply has been placed in the attachment.
Sincerely yours,
Prof. Yuhua Long,
School of Chemistry
South China Normal University
378 West Outer Ring Road, 510006, Guangzhou, P. R. China
Email: longyh@scnu.edu.cn

Reviewer 2 Report
Comments and Suggestions for Authors
1. Please provide justification for how this novel bioactive compound could be effectively used against pathogenic organisms, inflammation and oxidative stress.
2. The authors have investigated the novel bioactive compound against phytopathogenic and human pathogenic organisms, as well as its anti-inflammatory and oxidative stress potential. Please justify how these investigations are correlated and interconnected with this study. In my opinion, focusing on one particular problem or study would provide a better scientific solution. The authors should justify this approach.
3. The authors need to provide justification for how they identified the purified bioactive single fraction without impurities in TLC. They should also justify the selection of the solvent system used for this study. Additionally, if possible, please include related TLC photos in the supplementary documents to facilitate further purification of bioactive compounds.
4. The authors are advised to perform in silico modeling of a specific target or focus on a particular problem.
Comments on the Quality of English LanguageFurther refinement of English is needed.
Author Response

(The authors gave the same response as above.)

Reviewer 3 Report
Comments and Suggestions for Authors
The purpose of this paper is to isolate a new alkaloid named penifuranone A (1), as well as three known compounds (2-4) from the mangrove endophytic fungus Penicillium crustosum, and to evaluate their antimicrobial, antioxidant, and anti-inflammatory activities. Potential phytochemical and biomedical applications of this work can be foreseen, and it is generally recommended for publication with very few comments.
1. Structural identification is critical for the discovery of new bioactive ingredients, and some data are missing regarding the characterizations, such as UV-vis spectrometry and FT-IR, which are suggested to be supplemented. Additionally, information on yield and the purity of the newly isolated compound is needed.
2. In plant materials, some data are missing and recommended to be added, such as a picture of Penicillium crustosum, especially during the collection process, information about who collected and identified the plant, collection time, data regarding the place of collection, and voucher number of the plant.

Author Response

(The authors gave the same response as above.)

Reviewer 4 Report
Comments and Suggestions for Authors
This manuscript reports on the isolation of four compounds, one of them new alkaloid, from the mangrove endophytic fungus Penicillium crustosum, and their antibacterial and anti-inflammatory activity. The methods applied are up-to-date and appropriate, the conclusions are supported by the results. There are only a few minor concerns, as follows:
1. The Introduction is too wordy, should be written in a more concise way.
2. Concerning the semi-synthesis of compound 1a, the Authors should explain why they decided to synthesize the hexadecanoyl derivative, and not any other one.
3. Anti-inflammatory assay: Only compounds 1 and 2 were tested. Why weren’t the other compounds tested? Moreover, in the Conclusion the Authors claim “And all compounds were tested for anti-inflammatory, antioxidant and antimicrobial activities.”, which is not true
Comments on the Quality of English LanguageModerate editing of English language required
Author Response

(The authors gave the same response as above.)

Reviewer 5 Report
Comments and Suggestions for Authors
The study about novel alkaloid from fungus Penicillium crustosum is unique and well arranged. Only have minor queries-
1. Is there any taxonomic identification from plant Acanthus ilicifolius?
2. Please provide experiments conditions of HRMS data obtained by QTRAP.
3. Is there purity analysis of the compound? I assume that atleast the novel compound should be analyzed for purity either by TLC or LC methods.
4. Page 7, line 179, author mentioned about TLC, but I don’t see the TLC analysis in the study? Did you analyze the compounds 1-4 by TLC or only column fractions were analyzed. As far as I know alkaloids are best visible in dragendroff reagent. Did you use that or some other reagent for visualization?
Author Response

(The authors gave the same response as above.)

Round 2
Reviewer 5 Report
Comments and Suggestions for Authors
The authors well answered all the queries. I have some suggestions below before the final publication.
Page 1, Line 38, cite the reference in support of statement- https://doi.org/10.2174/1389201020666191015161429.
There are some errors in the punctutation marks in the MS. Kindly improve it with utmost care.
Make all references uniform.
Author Response

(The authors gave the same response as above.)
